# Effect of Multistage Solution Aging Heat Treatment on Mechanical Properties and Precipitated Phase Characteristics of High-Strength Toughened 7055 Alloy

**DOI:** 10.3390/ma17081754

**Published:** 2024-04-11

**Authors:** Qilun Li, Xiaobo Zhang, Ce Guo, Jisen Qiao

**Affiliations:** 1School of Materials Science and Engineering, Lanzhou University of Technology, Lanzhou 730050, China; 18215128251@163.com (Q.L.); gc113959@163.com (C.G.); 2State Key Laboratory of Advanced Processing and Recycling of Nonferrous Metals, Lanzhou University of Technology, Lanzhou 730050, China

**Keywords:** 7055 aluminum alloy, solution aging heat treatment, mechanical properties, precipitation characteristics

## Abstract

In this paper, a one-step hot extrusion dual-stage solution treatment method is employed to fabricate high-strength and tough T-shaped complex cross-section 7055 (Al-Zn-Mg-Cu-Zr) alloy profiles, and a detailed investigation is conducted on the microstructure and mechanical properties. The results indicate that the comprehensive mechanical properties of the 7055 aluminum extruded alloy using the two-stage solution aging treatment are excellent. This is particularly evident in the balance between strength and ductility, where outstanding strength is accompanied by a plasticity that is maintained at 13.2%. During the extrusion process, the deformation textures are mainly composed of brass and copper, forming a 15.1% recrystallization texture Cube. In addition, the equilibrium phase η(MgZn_2_) precipitated in the grain is the main strengthening phase, and there are large discontinuous grain boundary precipitates at the grain boundary, which hinders the grain boundary dislocation movement and has great influence on the mechanical properties of alloy materials.

## 1. Introduction

Al-Zn-Mg-Cu series (7XXX series) aluminum alloys find extensive application in the modern military, automotive transportation, and aerospace sectors for heavy structural components, owing to their exceptional properties, such as low density, ultra-high strength, high fracture toughness, excellent fatigue resistance, and stress corrosion resistance [1,2,3,4,5,6,7,8,9]. Since the beginning of the 21st century, spurred by the rapid advancements in the aerospace industry, the utilization rate of high-strength, high-toughness aluminum alloys in aircraft structure manufacturing has surpassed 90%. Consequently, the need for developing high-performance aerospace structural aluminum alloys has grown increasingly pressing, particularly concerning their processing capabilities, toughness, and corrosion resistance. Nevertheless, conventional processing and forming methods have often fallen short of meeting the rigorous standards of the aerospace industry [10,11,12]. In response to challenges like grain boundary segregation and reduced plasticity resulting from excessive alloying element doping and casting-induced oxidation inclusions, the hot extrusion processing technique, proposed by experts and scholars in recent years, has emerged as an effective means of shaping Al-Zn-Mg-Cu alloys.

Precipitation phases formed during the hot extrusion of aluminum alloys have a great influence on their microstructure and mechanical properties [13,14,15]. Thus, excellent mechanical properties can be obtained by optimizing the hot extrusion and solid solution aging process, which affect the morphology and distribution of the precipitated phases. Ren et al. [16] found that regression re-aging (RRA) can effectively strengthen 7055 alloy, which is mainly due to the presence of a GP region, η’ phase, and precipitation. Moreover, the research also indicated that the interaction between dislocation and precipitated phase was the main strengthening mechanism of 7XXX alloy. The strength mechanism of the alloy can be expressed by the following formula [17]:(1)$$\sigma =\sigma_{\mathrm{Al}}+\sigma_{\mathrm{dis}}+\sigma_{\mathrm{GB}}+\sigma_{\mathrm{sol}}+\sigma_{\mathrm{pre}}$$
where $\sigma_{\mathrm{Al}}$ is the strength of pure aluminum, which can be regarded as a constant. $\sigma_{\mathrm{dis}}$, $\sigma_{\mathrm{GB}}$, $\sigma_{\mathrm{sol}}$, $\sigma_{\mathrm{pre}}$ are the strengthening factors of the dislocation strengthening, grain boundary, solid solution, and precipitation phase, respectively. It is pointed out that injection-molded Al-Zn-Mg-Cu-Zr alloy after hot extrusion is mainly strengthened by its grain boundary and dislocation, while its strengthening after the aging process mainly occurs due to precipitation phase, such as MgZn_2_ and Al_3_Zr. Jian Ren et al. [18] highlighted that elevated levels of Zn and Mg contribute to enhancing the density of the strengthened phase in Al-Zn-Mg-Cu alloy. However, a reduced copper content can impede the precipitation of the S(Al_2_CuMg) phase, thereby facilitating the solid solution treatment process. The formation of superfine crystals and uniform secondary phases during extrusion results in exceptional mechanical properties, attributed to the combined strengthening mechanisms of grain boundary reinforcement and precipitation hardening. Xu et al. carried out multistage solution aging treatment on extruded Al-11.2Zn-3.0Mg-1.3Cu-0.2Zr aluminum alloy, and the results proved that with increases in the solution time and temperature, the strength of the alloy showed a trend of first increasing and then decreasing [19]. This is mainly because the size of precipitated phase coarsens with further aging, which reduces the dispersion degree of the precipitated phase, and thus, reduces the strength of the alloy.

Numerous international researchers have also extensively investigated the enhancement of properties of Al-Zn-Mg-Cu alloys. Sung-Jae Won et al. [20] explored the influence of Mg/Cu content on the microstructure and mechanical properties of these alloys. They found that altering the Mg/Cu ratio resulted in stronger solid solution strengthening effects, leading to an overall improvement in alloy performance, and thus, achieving the objective of alloy optimization. Diya Mukherjee et al. [21] aimed to investigate the variations in the mechanical properties of Al-Zn-Mg-Cu-X alloys under different heat treatment conditions through microalloying modification. The study results revealed that the addition of the Sc element led to the highest tensile strength of the alloy in the T6 state, attributed to the presence of fine dispersoids. Furthermore, after RRA treatment, the alloy’s elongation was enhanced without compromising its strength.

Achieving an optimal balance between enhanced strength and plastic toughness primarily involves grain size refinement [22,23]. While elevating the mass fractions of Mg and Zn can facilitate solid solution strengthening and precipitation hardening, excessive alloying elements may result in macroscopic segregation and diminish the alloy’s plasticity [24,25]. Furthermore, reducing work hardening and dislocation density during thermal deformation and heat treatment, coupled with the recrystallization process, enhances the alloy’s plasticity but tends to lower its strength. Thus, we propose employing a hot extrusion forming technology paired with a two-stage solid solution aging treatment to attain high strength while modestly increasing the plasticity. This approach is applicable to the preparation of extruded T-profile products endowed with excellent comprehensive mechanical properties, offering valuable insights for the design and manufacture of lightweight aerospace aluminum structures.

In conclusion, hot extrusion presents a cost-effective and streamlined one-step forming process in contrast to other molding techniques, thereby enhancing production efficiency. Hence, this study focuses on fabricating complex cross-sectional 7055 alloy T-profiles (Al-8.1Zn-2.05Mg-2.4Cu-0.12Zr-0.025Ti) through a combination of semi-continuous casting ingot micro-alloying and backward extrusion methods. The resulting alloy exhibits a favorable balance between strength and ductility, primarily attributed to the optimized distribution of precipitation phases. This research extensively characterizes the microstructural evolution and mechanical properties of the designed 7055 aluminum alloy during the extrusion process, offering detailed insights into its property enhancement.

## 2. Experimental Materials and Methods

The experimental material used in this paper was 7055 aluminum alloy; the main alloy composition was designed as Al-Zn-Mg-Cu-Zr. The reference chemical composition of 7055 aluminum alloy is given in Table 1 (all in wt.%). A T-shaped material with a complex cross-section was prepared using a combination of semi-continuous ingot casting and one-step extrusion forming. Subsequently, the extruded profile underwent a two-stage solution treatment, followed by artificial aging to obtain the T-shaped material used in the experiment. As shown in Figure 1, the alloy ingots for semi-continuous casting were subjected to a 455~465 °C/24 h homogenization heat treatment. The extruded 7055 alloy was solid solution-treated at 470 °C/3 h, followed by quenching in water at 35–40 °C with a transition time of less than 15 s. Subsequently, it was subjected to a double-stage artificial aging treatment. The aging process was 121 °C/5 h + 155 °C/7 h.

Before conducting tensile and compressive performance tests, the test specimens were prepared by cutting aluminum alloy ingots, heat-treated ingots, and extruded T-shaped materials subjected to solution treatment and aging treatment in the extrusion direction. The specimens were sandpapered smooth and then subjected to a room temperature tensile and compression test on a Shimadzu Autograph AGS-X tester (Tokyo, Japan) with a constant strain rate of 1 × 10^−3^ s^−1^. To ensure the accuracy of the alloy performance testing, we also assessed the strength and elongation of the alloy at different strain rates. The dimensions of the experimental specimens are shown in Figure 2; the original specimens were 10 mm in length and 1.5 mm in thickness. To obtain reliable experimental results, accurate engineering stress–strain curves and the average ultimate tensile strength, yield strength, and total elongation after fracture were finally obtained by three parallel experiments.

The physical phases of the 7055 alloy ingots and extruded profile specimens were characterized by X-ray diffractometer (XRD) combined with monochromatic Cu target radiation with a scan rate of 4°/min and a scan range 2 θ of 15°~85°. The microstructures of the alloy specimens were characterized using a Zeiss LSM800 laser (Jena, Germany) scanning confocal microscope, a FEG-450 scanning electron microscope (SEM) and its associated EDS equipment, electron backscatter diffraction (EBSD), and the FEI Talos F200 field emission transmission electron microscope (TEM). Specimens for optical microscope (OM) observation were mechanically polished and then etched with Keller’s reagent. SEM samples were cut along the extrusion direction (ED) and mechanically polished. Specimens with extruded profiles for TEM inspection were sliced along the extrusion direction, then mechanically polished to 70 μm and electrolytically polished with a 10% perchloric acid solution in a double jet at −25 °C.

Theoretical analysis of the strengthening mechanisms present in the employment of Al-Zn-Mg-Cu alloy for extruding T-profiles in 7055 aluminum alloy was carried out. In the case of Al-Zn-Mg-Cu alloy, four primary strengthening mechanisms prevail: solid solution strengthening, dislocation strengthening, grain boundary strengthening, and precipitation strengthening [23,26].

Among these, grain boundary strengthening is calculated according to the Hall–Petch equation [27,28,29]:(2)$$\sigma_{\mathrm{GB}}=\sigma_{0}+k_{y}/\sqrt{d}$$
where *σ*_0_ is the yield stress; *σ*_0_ is the onset stress of dislocation motion; *k_y_* is the strengthening factor, which is about 0.22 MPa for the Al-Zn-Mg-Cu alloy [23]; and *d* is the mean grain size.

According to the Orowan cycle mechanism, the precipitation enhancement (Δ*σ_ppt_*) is calculated using the Orowan equation [30,31]:(3)$$∆\sigma_{\mathrm{ppt}}=M(0.4\mathrm{Gb}/\pi \sqrt{1-\nu })\left[(\ln2\bar{r}/b)/\lambda_{p}\right]$$
where *M* is the average orientation coefficient, and aluminum alloys are face-centered cubic (f.c.c) polycrystals with an *M* of 3.06; *G* is the shear modulus, which is 26.9 GPa for 7055 aluminum alloy; *b* is the magnitude of Burgers vector (0.286 nm for Al); *υ* is the Poisson’s ratio of aluminum, which is 0.33; $\bar{r}\;$is the circular cross-section of the spherical precipitates in a random plane average radius, $\bar{r}=\sqrt{2/3}r$; *r* is the average radius of the precipitated phase; *λ_p_* is the precipitate spacing between edges, which can be roughly estimated using the following relation [23]:(4)$$\lambda_{p}=2\bar{r}\left(\sqrt{\frac{\pi }{4f}}-1\right)$$
where $\bar{r}\;$is the circular cross-section of the spherical precipitates in a random plane average radius, $\bar{r}=\sqrt{2/3}r$; *r* is the average radius of the precipitated phase; *f* represents the volume fraction of precipitates.

The strengthening increment caused by dislocations can be described as [32]:(5)$$∆\sigma_{\mathrm{dis}}=M\alpha \mathrm{Gb}\rho^{1/2}$$
where *M* is the Taylor factor, *α* is a constant around 0.35, *G* is the shear modulus, *b* is the magnitude of Burgers vector (0.286 nm for Al), and *ρ* denotes the dislocation density.

The solid solution strengthening effect can be governed by [33]:(6)$$∆\sigma_{\mathrm{ss}}=\mathrm{MGb}\varepsilon_{ss}^{3/2}\sqrt{c}$$
where $\varepsilon_{\mathrm{ss}}$ = 0.38 is an experimental constant; *M* is the average orientation coefficient, and aluminum alloys are face-centered cubic (f.c.c) polycrystals with an *M* of 3.06; *G* is the shear modulus, which is 26.9 GPa for 7055 aluminum alloy; *b* is the magnitude of Burgers vector (0.286 nm for Al); and *c* is the solute concentration (wt.%).

## 3. Results

### 3.1. Mechanical Properties

Figure 3 shows the engineering stress–strain curves (Figure 3a,c,d) and microhardness histograms (Figure 3b) for the quasi-static tensile and compressive testing of the 7055 alloy ingots, heat-treated ingots, and extruded profiles subjected to solution treatment and aging. The results of the mechanical properties of the alloy show that the strength of the extruded 7055 alloy T-profile after solid solution treatment was greatly increased. The ultimate tensile strength reached 650 MPa, and it also demonstrated excellent compression performance. The tensile and compressive strengths at different strain rates were evaluated, and the best overall performance of the alloys was identified at a strain rate of 1 × 10^−3 ^s^−1^. When compared to the alloy before extrusion, significant improvements were observed in both the tensile strength and elongation, with a notable increase in the maximum elongation, which reached 13.2%. Regarding the hardness change, the hardness of the 7055 profile after the solid solution treatment after extrusion reached 190.4 HV, while for the aluminum alloy in the as-cast state, its hardness was lower after the homogenizing treatment than before the heat treatment.

After obtaining the tensile properties of 7055Al with a squeezed T-shaped cross-section, a comparison was made with the performance of Al-Zn-Mg-Cu series alloys with equivalent compositions as reported in the reference literature (e.g., Figure 4). Figure 4a depicts the mechanical parameters of the alloy obtained in our study. It is evident in the figure that the 7055 aluminum alloy extruded into T-profiles and subjected to two-stage solution aging treatment exhibit excellent comprehensive performance. Figure 4b compares the tensile properties of the 7055 alloy with those of other Al-Zn-Mg-Cu alloys prepared using different methods. As observed in the graph, the T-profiled materials fabricated using our strategy of semi-continuous casting ingots combined with reverse extrusion show improved strength without sacrificing ductility, surpassing the mechanical properties reported for most previously studied Al-Zn-Mg-Cu alloys.

The SEM analysis of the tensile test revealed the fracture morphology of the tensile sample, as depicted in Figure 5. In examining the fracture morphology of the extruded profile (Figure 5a,d), numerous dimples and tearing edges within the grain were observed, along with secondary precipitated particles S (Al_2_CuMg) and impurity particles containing Ti in the dimples. The fracture exhibited characteristics of ductile fracture combined with dimples and slip bands, indicating that the fracture process is mainly controlled by the ductility behavior associated with dislocation activities. Figure 5b,e and Figure 5c,f, respectively, illustrates the tensile fracture morphology of the ingot with homogenized fire treatment and without heat treatment. The fractures displayed a rock sugar block, cleavage table, and river pattern. Consequently, it was determined that the fracture types of the ingot included intergranular fracture, cleavage fracture, and phase junction mixed fracture. It is noteworthy that after homogenization, the grain size of the 7055 aluminum alloy decreased, and the cleavage fracture became less pronounced, which is also a significant factor contributing to the poor plasticity of the ingot material.

### 3.2. Microstructural Analysis

Figure 6 shows the metallographic microstructures of the 7055 aluminum alloy homogenized ingot and the solid solution aging-treated extruded T-profile. In Figure 6a,b, it can be seen that the 7055 alloy after equalizing treatment has coarse equiaxial grains, with a large number of eutectic phases and grain boundary phases inside the grains and at the grain boundaries. Figure 6c,d shows that the extruded heat-treated profile became obviously fine in microstructure, with a dense distribution of the grain boundaries and an obvious improvement in grain boundary segregation. Additionally, the grains are typically fibrous along the extrusion direction. The distribution of grains along the extrusion direction is typical fibrous, and there is some reversion and recrystallization.

To further analyze the phase transformation characteristics of the specimens before and after extrusion, the phase structure and composition of the specimens were analyzed by XRD and EDS. The XRD patterns of the as-cast and extruded 7055 T-profiles are shown in Figure 7a,b, respectively. There are diffraction peaks of α-Al matrix phase and weak η(MgZn_2_) phase in the as-cast samples. In addition, diffraction peaks of the α-Al matrix phase and η(MgZn_2_) phase also exist in the extruded profile sample. This indicates that the η phase in the 7055 alloy grew and coarsened during the extrusion and solution aging treatment, forming more stable precipitation strengthening phase, which is consistent with the TEM observation results. It was shown that the precipitation of the η’ and η phases definitely occurred in 7055 alloy after a solid solution treatment [37,40]. Ma et al. described their study of the relationships among precipitation phenomena, grain size, and mechanical behavior in a complex precipitation-reinforced alloy system of 7075Al alloy. Furthermore, the strengthening mechanism of the alloy was analyzed in detail [23].

Figure 8 shows the SEM micrographs and EDS surface scan distribution of the as-cast and extruded profile samples. It can be seen that a large number of grain boundary phases were distributed at the grain boundaries in the as-cast state (in Figure 8a). Analysis of the EDS point scan results indicates that the grain boundary phases consisted of Cu, Mg, and Zn. In contrast, the microstructure of the extruded specimens changed significantly (in Figure 8b), and the white phases in the matrix were significantly decreased in size and distributed in chains or strips at the grain boundary area, which mainly consisted of Mg, Zn, and Zr. This shows that the alloying elements were almost completely dissolved in the matrix during the extrusion and solid solution treatment, while the Al_7_Cu_2_Fe eutectic phase was precipitated at the grain boundaries. Al_7_Cu_2_Fe was present in the study by Liu et al., and their results demonstrated that it also had an effect on the properties of the alloy [40].

EBSD was used to determine the effect of the double-stage solid solution aging treatment on the microstructure and recrystallization of the material after extrusion. Figure 9 shows the EBSD plots of the aluminum alloy extruded profile at high magnification after the double-stage solid solution aging treatment. Figure 9a displays an inverse polar diagram (IPF) of the 7055 alloy extruded profile sample, in which it can clearly be seen the grains of the 7055 aluminum alloy extruded profile are dominated by the typical elongated grains, which are arranged longitudinally along the extrusion direction. Some of the grains are equiaxed, which is due to recrystallization during the solid solution aging treatment. The grain boundary orientation distribution shows that there are a large numbers of small-angle grain boundaries and sub-grains inside the extruded profile. The difference in the grain boundary orientation angle is less than 15°, of which the small-angle grain boundaries account for 50.1%, and in which some of the recrystallization organization appear. The size of the grains in the figure varies greatly, and there are fine grains embedded at the grain boundaries of the large elongated grains. The statistics of the grain size (Figure 9c) show that the average grain size of the extruded profile was 5.18 μm, while the majority of the grain sizes were concentrated around 2~5 μm, as observed in the frequency distribution curve, which shows that the alloy grains were obviously refined after extrusion.

Figure 9b shows the grain orientation spread (G.O.S.) plot, which has a low G.O.S. value due to the absence of strain in the recrystallized grains. The purple area in the figure shows the recrystallized grains, and it can be seen that most of the grains did not recrystallize and the degree of recrystallization was relatively low. It is worth noting that, combined with the statistical distribution of the grain boundary orientation difference (Figure 9), the low-angle grain boundaries (LAGBs) occupied a certain proportion; specifically, the frequency of grain boundaries around 2° reached 0.7, indicating that there were a large number of deformed grains inside the alloy material, which made the dislocation density increase. Meanwhile, the presence of sub-grain boundaries (2~15°) and various degrees of high-angle grain boundaries (HAGBs) absorbed a certain amount of dislocations, which led to a decrease in the dislocation density.

The orientation distribution function (ODF) plot shown in Figure 10 illustrates the density of the grain orientation distribution, and the relevant main weave components and their volume fractions were calculated, as shown in Table 2. In the 7055 aluminum alloy extruded profile, the main deformation weaves produced are classified as copper {110}<111> and brass R{111}<112> and {552}<115> [41]. From the plot, it is evident that the strongest orientation density lies along <111>, indicating the preferential growth of the alloy grains in the <111> direction during the extrusion and heat treatment, resulting in this directional anisotropy. In addition, a recrystallized cubic weave Cube {001}<100> is also present. The weaker cubic weave is thought to favor the formability of the material [7,42]. Thus, the formation of the Cube texture significantly influences the material’s formability, enabling the one-step extrusion process to produce T-shaped profiles with outstanding performance, which highlights a crucial factor behind their excellent properties.

### 3.3. Precipitation Characteristics

Figure 11 shows a micrograph of the grain boundary precipitation phase particles of the 7055 alloy extruded profile. The different morphologies of the precipitated phases can be clearly observed at the grain boundaries. In Figure 11a,b, the precipitated phases are ellipsoidal and distributed along the grain boundaries in a chain shape, with an average particle size of 33 nm, and a precipitate-free zone (PFZ) can be observed at the grain boundaries. Figure 11c shows the grain boundary precipitated phase particles, with an average particle size of 167 nm, are completely separated and unevenly distributed at an angle to the grain boundaries. The presence of these large precipitates at the grain boundaries is attributed to prolonged low-temperature artificial aging. The coarse precipitates exhibit a discontinuous distribution along the grain boundaries, creating a precipitate-free zone (PFZ), with a width of 62 nm.

As shown in Figure 12, it can be observed that large quantities of dense, fine slate-like, and ellipsoidal nanometer-sized second phase particles were uniformly distributed inside the matrix.In SADP, it can be observed that in the [0$\bar{11}$]_Al_ orientation region, there are faint separated spots (marked by red circles) along 1/3 and 2/3{0$\bar{2}$2}_Al_ near the Al matrix diffraction spot, and the results indicate that this spot is η′ phase. In addition, the presence of Al_3_Zr was observed at 1/2{0$\bar{2}$2}_Al_, which could inhibit the occurrence of recrystallization and act as a nucleation core for the η phase, promoting precipitation [43].

Figure 13 presents clearer images of the precipitate phase morphologies, accompanied by rapid Fourier transformation to calibrate the diffraction spots in different distribution directions. This enables better observation of the size and distribution characteristics of the precipitate phases. In the figure, it can be observed that the growth directions and sizes of the precipitate phases are not consistent, but they are fine and dispersed, contributing significantly to the strengthening effect of the alloy. Figure 13c,d depicts the diffraction patterns and spots of the η phase, further confirming that the precipitate phase observed in the 7055 alloy extruded profiles is indeed η phase [16,17,27]. Figure 14 represents high-resolution TEM images of the precipitate phase in Figure 13b. Fast Fourier transform (FFT) and inverse Fourier transform (IFFT) were used to determine the type of precipitated phases and their dislocation relationship. These nanoscale precipitates do not perfectly align with the Al matrix. In Figure 14(a1), the crystal plane spacing of the Al matrix along the ($\bar{1}1\bar{1}$)_Al_ direction was measured to be 0.225 nm, and the average length diameter of the precipitated phase was 27 nm. As shown in Figure 14(a1), η phase is semi-coherent or non-coherent with the Al matrix due to obvious atomic dislocations and lattice distortion [30].

## 4. Discussion

### 4.1. Strengthening Mechanisms

In this study, complex cross-sectional 7055 aluminum alloy shaped materials were prepared by melting semi-continuous casting ingots combined with hot extrusion forming. During the processes of the extrusion molding and heat treatment, the enhancement in the strength of the alloy arose from factors such as work hardening due to plastic deformation, high-density dislocations, and the temperature and duration parameters of the heat treatment. These augmented strength values stemmed from the cumulative effects of various strengthening mechanisms. The primary strengthening mechanisms influencing the alloy’s strength in this study are grain boundary strengthening, resulting from extrusion plastic deformation, and precipitation strengthening, caused by the widespread distribution of precipitates.

The average grain size of the 7055 cast alloy was 130.7 μm, while in the extruded profile, it can be seen in Figure 9 that the grain size was significantly refined, with an average grain size of 5.18μm. Therefore, it can be seen in Formula (2) that the smaller the grain size is, the more significant the alloy strengthening effect is, which also indicates that grain refinement during extrusion is an important reason for improving the strength of extruded profiles.

With the degree of aging, the precipitation sequence of the precipitated phases is α supersaturated solid solution → GP(I, II) zone → η’ phase → η(MgZn_2_) phase [44,45,46,47]. According to Equations (3) and (4), it can be inferred that under the conditions of a smaller precipitate size and a simultaneously larger volume fraction, the strengthening effect due to precipitation becomes more pronounced. In this study, the presence of η(MgZn_2_) phase was observed in both the Al matrix and grain boundaries in the 7055 aluminum alloy extruded profiles (shown in Figure 11), and the η(MgZn_2_) phase was diffusely distributed inside the matrix. The statistics of the precipitation phase size of the intracrystalline dispersion can be seen in Figure 15; the average grain size of the precipitation phase reached 27 nm. According to the frequency distribution curve, it can be seen that the grain size of most of the precipitated phase was about 37.08 nm, which is an important factor for maintaining the high level of strength of the 7055 alloy extruded profile.

In addition, the grain boundary precipitation was coarse compared to the intracrystalline precipitation phase due to the fact that grain boundaries are face defects with higher energy and much higher driving force for both nucleation and growth than the intracrystalline. At the same time, the fast diffusion of atoms at the grain boundaries and the slow diffusion of solute atoms within the grain lead to the depletion of solute atoms on both sides of the grain boundaries, resulting in an apparently precipitation-free zone at the grain boundaries. Moreover, the precipitation phase of the large-angle grain boundary and the grain boundary precipitation-free zone are coarser and wider than that of the grain boundary.

### 4.2. Effect of Extrusion Textures on Mechanical Properties

It has been shown that in extruded aluminum alloys, the main existing rolling weaves are classified as S, brass, and copper, while the main components of recrystallized weaves are rotating Goss, Goss and Cube. More importantly, the different orientations formed during extrusion weaken the deformation weaves, which affects the mechanical properties of the material [48,49]. Due to the large plastic deformation of 7055 aluminum alloy during the extrusion process, the grain refinement of the alloy is significant (as shown in Figure 9), and recrystallization occurred during the solution treatment. The plastic deformation of the material will affect the grain orientation and microstructure of the alloy, which has a significant impact on the mechanical properties of extruded aluminum alloy [6,50]. The deformation texture of the face-centered cubic crystal mainly changed from the <100> to the <111> and <101> directions, and a large amount of copper {110}<111> texture produced in the 7055 aluminum alloy during extrusion was transformed due to deformation. It can be seen that the textures generated in the extrusion process were mainly distributed in the extrusion direction, and the macroscopic mechanical properties show obvious anisotropy, which makes alloys exhibit excellent strength and toughness in the extrusion direction [41]. Consequently, the weaving created during extrusion is an important reason for the increased strength and toughness of extruded profiles.

## 5. Conclusions

The present study employed a method combining semi-continuous casting ingots with reverse extrusion to fabricate T-profiled materials using 7055 aluminum alloy for aerospace applications. The research investigated the microstructure and mechanical properties of the extruded profiles after a two-stage solid solution aging treatment and discussed the characteristics of the precipitated phases in this condition. The main conclusions were obtained as follows:(1)The alloy T-profile after extrusion exhibited excellent mechanical properties, and its ultra-high strength was mainly due to the fine grain size, high density, and high volume fraction of intracrystalline precipitation phase. The alloy tensile strength and elongation reached 650 MP and 13.2%, respectively.(2)The fibrous and deformed organization of the extruded 7055 aluminum alloy T-profile exhibited a copper {110}<111> deformation weave, a dominating {100}<001> recrystallized cubic weave, and selective grain growth in the extrusion direction, which caused anisotropy in the alloy. In addition, the work hardening caused by severe plastic deformation during the extrusion played an important role in the improvement in the mechanical properties of the 7055 alloy.(3)A high density of dislocations was produced in the 7055 aluminum alloy extruded profiles due to severe plastic deformation. The main strengthening phase present in the grain interior and at the grain boundary was the equilibrium η-phase, with an average grain size of 26.54 nm. The contribution of precipitation strengthening for the 7055 alloy extrusion T-profile was greater than that of the grain boundary strengthening, so the strengthening mechanism in the extrusion profile alloy treated by two-stage solid solution aging is mainly precipitation strengthening with the interaction of dislocation and precipitation phase.

## Figures and Tables

**Figure 1 materials-17-01754-f001:**
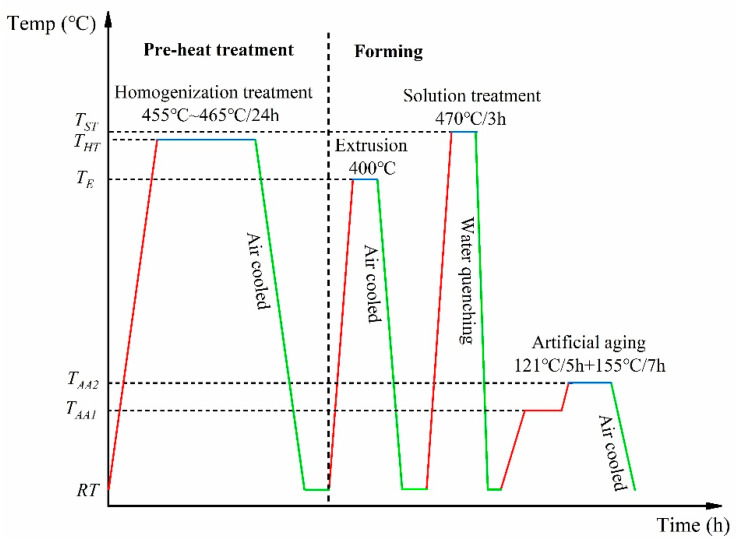
Temperature distribution of forming and heat treatment processes. (The red line indicates the temperature increase during heat treatment; The blue line indicates that processing is in progress; The green line indicates a decrease in temperature).

**Figure 2 materials-17-01754-f002:**
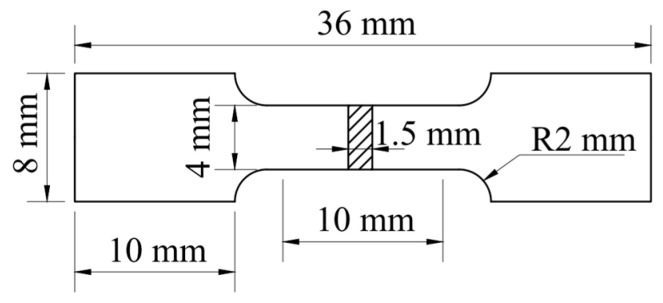
Dimensions of tensile samples.

**Figure 3 materials-17-01754-f003:**
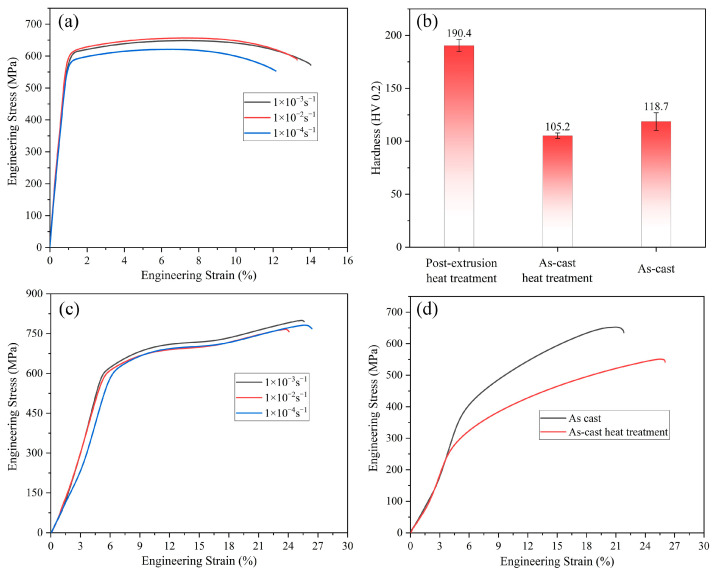
Mechanical property curves of 7055 alloy: (**a**) tensile strength curves at different strain rates; (**b**) microhardness curves; (**c**) compressive strength curve at different strain rates; (**d**) compressive strength curve of 7055 alloy as cast.

**Figure 4 materials-17-01754-f004:**
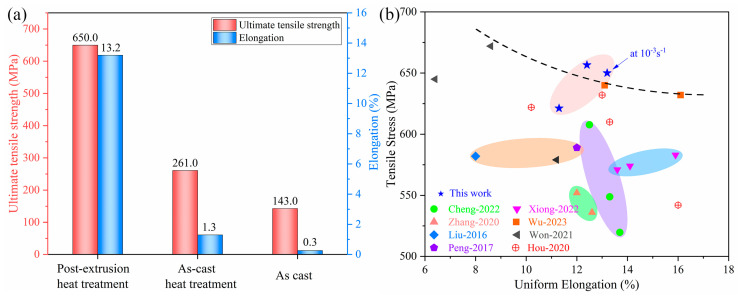
Performance of 7055 alloy: (**a**) strength-ductility of 7055 alloy before and after extrusion and heat treatment; (**b**) comparison of tensile strength and ductility for Al-Zn-Mg-Cu series alloys [20,26,34,35,36,37,38,39].

**Figure 5 materials-17-01754-f005:**
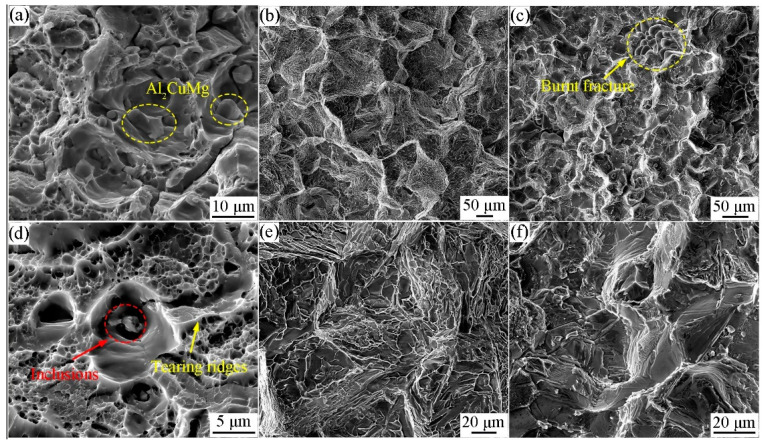
SEM fracture morphology images of 7055 aluminum alloy in the conditions of (**a**,**d**) extruded profiles, (**b**,**e**) heat-treated ingot, and (**c**,**f**) as-cast.

**Figure 6 materials-17-01754-f006:**
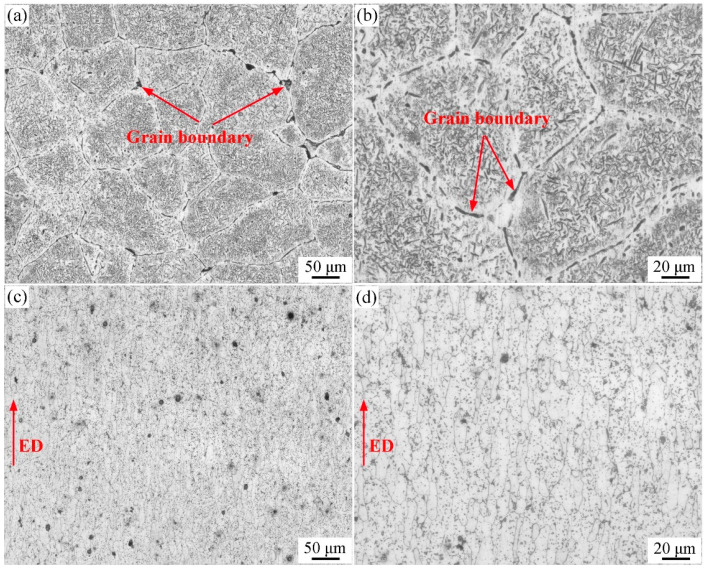
Optical micrographs of 7055 aluminum alloy samples in various states: (**a**,**b**) as-cast; (**c**,**d**) extruded profile.

**Figure 7 materials-17-01754-f007:**
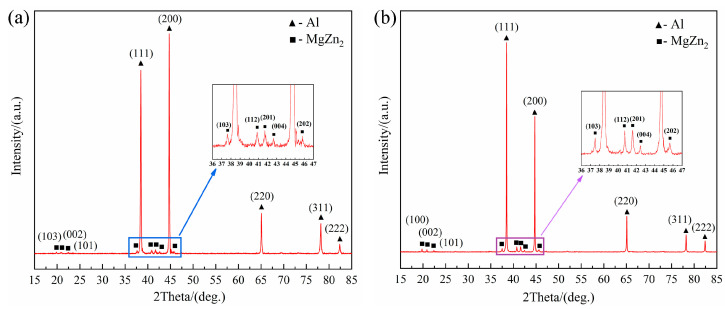
XRD patterns of the alloy in various states: (**a**) as-cast; (**b**) extruded profile.

**Figure 8 materials-17-01754-f008:**
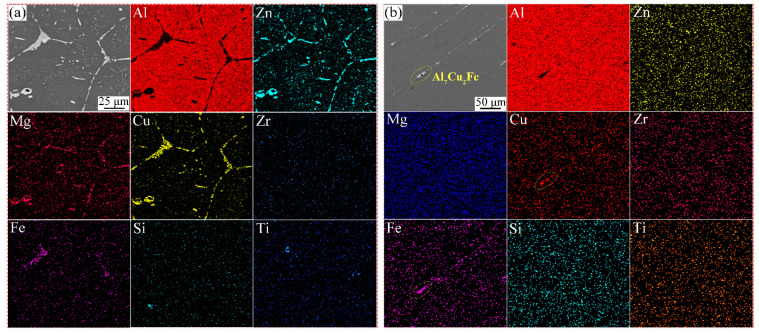
SEM images of 7055 aluminum alloy in various states: (**a**) as-cast; (**b**) extruded profile.

**Figure 9 materials-17-01754-f009:**
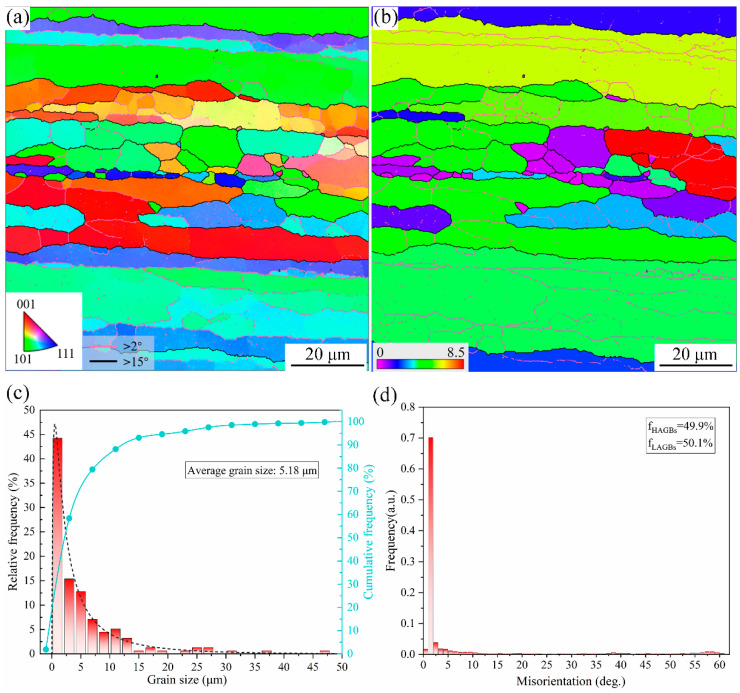
EBSD of 7055 aluminum alloy extruded profile specimen: (**a**) IPF map; (**b**) grain orientation spread (G.O.S.); (**c**) statistical graph of grain size; and (**d**) statistical map of grain boundary misorientation.

**Figure 10 materials-17-01754-f010:**
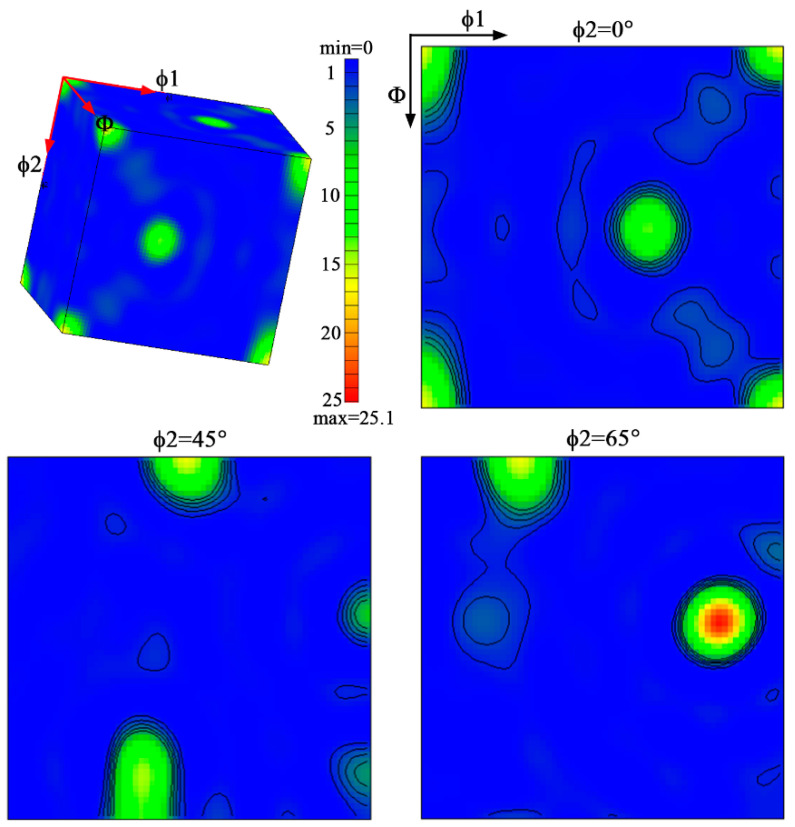
Orientation distribution function (ODF) images of 7055 aluminum alloy extruded profile.

**Figure 11 materials-17-01754-f011:**
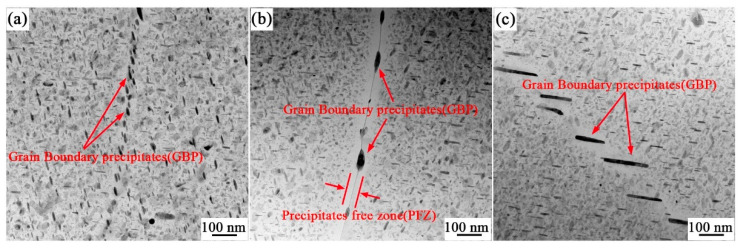
TEM images of 7055 aluminum alloy extruded profile sample. (**a**) Morphology of precipitated phase distributed continuously at grain boundaries; (**b**) Precipitation free zone; (**c**) Unevenly distributed strip precipitates at grain boundaries.

**Figure 12 materials-17-01754-f012:**
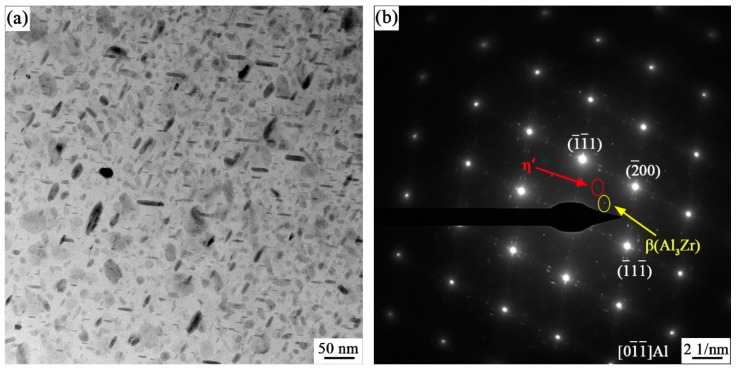
(**a**) TEM microstructure and (**b**) selected electron diffraction pattern of 7055 aluminum alloy extruded profile along [0$\bar{11}$]_Al_ orientation.

**Figure 13 materials-17-01754-f013:**
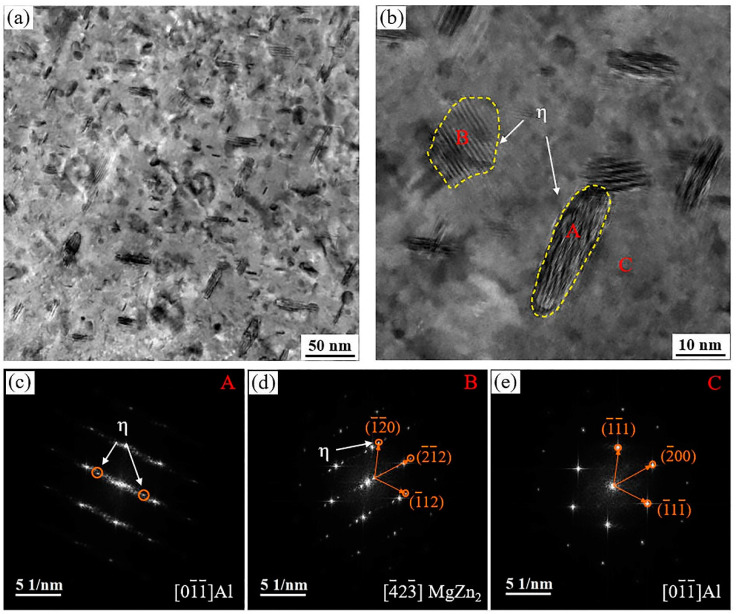
TEM images of 7055 aluminum alloy extruded profile: (**a**,**b**) images of the precipitation phase distribution inside the grain; (**c**–**e**) corresponding FFT images of A, B, C areas in (**b**). Here A, B, and C represent the η phase and Al matrix, respectively.

**Figure 14 materials-17-01754-f014:**
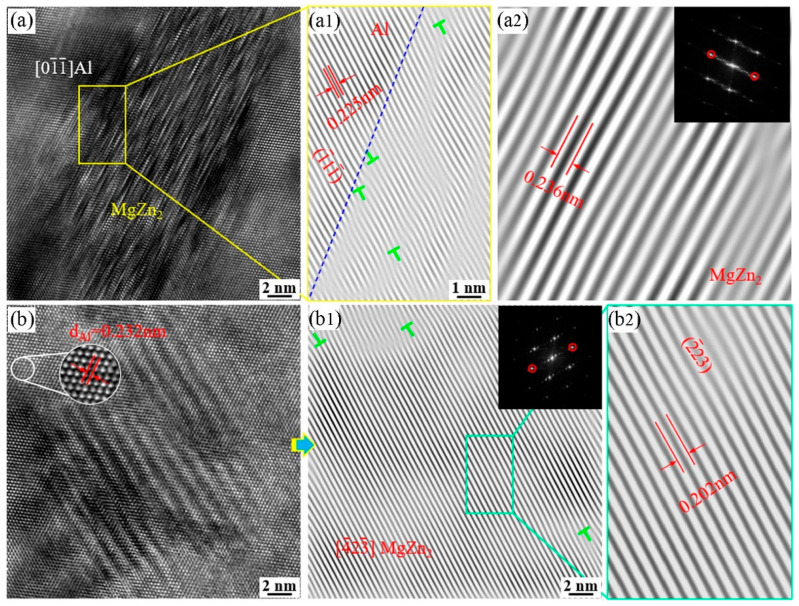
HRTEM images of 7055 aluminum alloy extruded profile: (**a**,**b**) HRTEM images of MgZn_2_ precipitated phases with different orientation distributions; (**a1**,**a2**,**b1**,**b2**) their IFFT images, respectively.

**Figure 15 materials-17-01754-f015:**
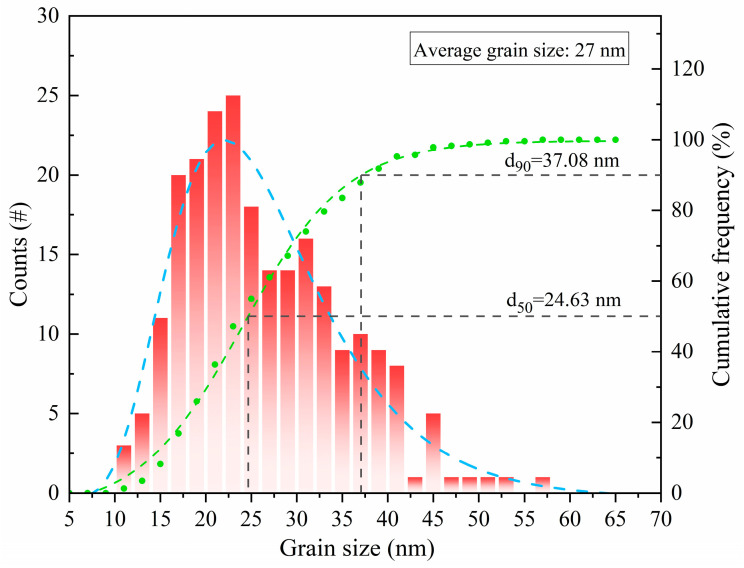
Statistical diagram of precipitate particle size.

**Table 1 materials-17-01754-t001:** Chemical composition of experimental materials.

Element	Zn	Cu	Mg	Zr	Ti	Al
wt.%	8.1	2.4	2.05	0.12	0.025	Bal.

**Table 2 materials-17-01754-t002:** Texture components of 7055 extruded profile sample.

Texture Components	Miller Indices	Fraction (%)
Cube	{001}<100>	15.1
Copper	{110}<111>	45.7
Brass R	{111}<112>	18.9
Deformation texture	{552}<115>	8.89

## Data Availability

The datasets generated during and/or analyzed during the current study are available from the corresponding author upon reasonable request.

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
