# Peer review of "Effect of Multistage Solution Aging Heat Treatment on Mechanical Properties and Precipitated Phase Characteristics of High-Strength Toughened 7055 Alloy"

_materials, 2024, doi:10.3390/ma17081754_

Round 1

Reviewer 1 Report

Comments and Suggestions for Authors

Table 1 is of no real utility; the composition of the alloy is already given in the text.

Figure 1 should contain on [Oy axis the actual values for temperature, not correspondence with the process (which the reader can deduce on their own).

What is the reasoning behind the successive steps the alloy was subjected to? How was the temperature chosen?

How does the strain rate correlate to actual strain the material will be subjected to in aerospace industry?

The results are very close for 10-3 and 10-2 s-1. How was the optimum chosen?

How do authors explain the different relative intensities of (111) and (200) reflections on Al (XRD data)?

Several aspects are discussed at a distance from the cited Figure  (see page 11, where after Fig 11 we see referenced Fig 15 without even a mention of Fig 13 and Fig 14).

Line 380: tissue evolution?

The novelty of the presented extrusion process results is not clear. How can these findings benefit the end user?

What is the energy cost and overall added cost of implementing this processing strategy of 7XXX alloys vs current methodologies?

The whole draft seems like putting together a series of results obtained through different characterization methods, but the data is not organized properly.

Comments on the Quality of English Language

Careful proofreading recommended.

Reviewer 2 Report

Comments and Suggestions for Authors

Title: “Effect of Multistage Solution Aging Heat Treatment on Mechanical Properties and Precipitated Phase Characteristics of High-Strength Toughened 7055 Alloy”

In this work the authors described a one-step hot extrusion dual-stage solution treatment method to fabricate high-strength and tough T-shaped complex cross-section 7055 alloy profiles. In particular, a detailed investigation on their microstructure and mechanical properties was conducted. The authors claim very good comprehensive mechanical properties of 7055 aluminum extruded alloy profiles after two-stage solution ageing treatment. This was shown through the balance between strength and ductility, since high strength was accompanied by a plasticity that is maintained at 13.2%. During the extrusion process, deformation textures were mainly composed of Brass and Copper, and formed 15.1% recrystallization texture Cube. In addition, the equilibrium phase η(MgZn2) precipitated in the grain was the main strengthening phase, and there were large discontinuous grain boundary precipitates at the grain boundary, which hindered the grain boundary dislocation movement and had a great influence on the mechanical properties of alloy materials.

General comment: This work should be deeply reworked to improve its quality and impact. In particular, the standard sections of a scientific contribution seem to be mixed in some parts of the current version of the main text. Some sections (e.g. “Introduction”) should be enlarged to better describe the current state of the art. Some figures should be reworked and improved together with their captions. In general, the quality of the language should be improved to allow the readers to better follow the logic flow of the main text.

Some detailed comments:

*) “1. Introduction “ section: This sections should be enlarged and a more comprehensive list of work should be provided to present a more detailed state of the art at the international level.

Lines: “Figure 3. Mechanical properties curves of 7055 alloy: (a) tensile strength curves at different strain rates, (b) microhardness curves, (c) compressive strength curve at different strain rates, (d) compressive strength curve of 7055 alloy as cast.

*) Figure 3 should be improved and the stain rates should be better described within the labels. More specifically, it is not clear whether the curve was totally experimental or it was a fit of some experimental points. Please explain better and improve.

Lines: “Figure 4. Performance of 7055 alloy: (a) Strength-ductility of 7055 alloy before and after extrusion and heat treatment; (b) Comparison of tensile strength and ductility for Al-Zn-Mg-Cu series alloys [20-24].”

*) Figure 4 should be improved. In particular, the background of figures should be avoided (better transparent). The (b) figure should be better explained. Please reworked

Lines: “Figure 8. SEM images of 7055 aluminum alloy in condition of (a) as cast and (b) extruded profile. “

*) The quality of these images should be improved, In particular, some subfigures are too dark to be correctly understood. Please correct.

Lines: “Figure 9. EBSD of 7055 aluminum alloy extruded profile specimen: (a) IPF map, (b) GOS, (c) statistical graph of grain size and (d) statistical map of grain boundary misorientation.

*) This figure should be improved together with its caption. Please do not use abbreviations only.

Perhaps a white background could be better.

Lines. “Figure 10. Orientation distribution function(ODF) images of 7055 aluminum alloy extruded profile.”

*) This figure should be better commented to the interested readers.

Lines: “Figure 13. TEM images of 7055 aluminum alloy extruded profile: (a)-(b) are the images of the precipitation phase distribution inside the grain, (c–e) the corresponding FFT images of A, B, C areas in (b). Here A, B and C represent η phase and Al matrix; respectively.

*) This figure should be improved and better explained.

Section: “4. Discussion “

*) This section should be reworked and improved. More specifically, all the formulas should be moved within the “Materials and Methods” together with their explanation.

References

*) A more comprehensive list of international works should be provided.

Comments on the Quality of English Language

The quality of the language should be improved

Reviewer 3 Report

Comments and Suggestions for Authors

The article discusses the microstructural and morphological features of high-strength 7055 alloy, the interest in which is due to its strength properties. In general, the presented line of research is very interesting and promising, and the work itself deserves attention and can be accepted for publication after the authors answer a number of questions that the reviewer had while reading it.

1. In the abstract, in addition to mentioning the name of the alloy, its elemental composition or chemical formula of the compounds from which it consists should be given, since many readers may not know the specific abbreviation for numbering alloys.

2. The presented alloy manufacturing scheme in Figure 1 contains quite a lot of processes associated with heating and cooling of the alloy, which leads to the formation of a complexly deformed structure. For clarity, authors are encouraged to use different colors to represent the heating and cooling processes to better illustrate the main steps.

3. When describing the hardness values obtained for various alloys, authors are recommended to consider such an effect as hardening and softening in order to show the greatest effect.

4. The authors should provide the percentage of the η(MgZn2) phase, which is formed as a result of deformation processes.

5. The authors point out that the formation of grains in the form of η(MgZn2) are a kind of strengthening grains, however, to confirm this, the authors should make a comparison with literature data, since such effects can also be associated with dislocation strengthening caused by texture effects.

6. The authors should explain in more detail the reasons for the formation of the Cube texture during deformation processes, what is this connected with and what is the root cause of such a transformation?

Round 2

Reviewer 2 Report

Comments and Suggestions for Authors

Title: “Effect of Multistage Solution Aging Heat Treatment on Mechanical Properties and Precipitated Phase Characteristics of High-Strength Toughened 7055 Alloy”

In this work the authors described a one-step hot extrusion dual-stage solution treatment method to fabricate high-strength and tough T-shaped complex cross-section 7055 alloy profiles. In particular, a detailed investigation on their microstructure and mechanical properties was conducted. The authors claim very good comprehensive mechanical properties of 7055 aluminum extruded alloy profiles after two-stage solution ageing treatment. This was shown through the balance between strength and ductility, since high strength was accompanied by a plasticity that is maintained at 13.2%. During the extrusion process, deformation textures were mainly composed of Brass and Copper, and formed 15.1% recrystallization texture Cube. In addition, the equilibrium phase η(MgZn2) precipitated in the grain was the main strengthening phase, and there were large discontinuous grain boundary precipitates at the grain boundary, which hindered the grain boundary dislocation movement and had a great influence on the mechanical properties of alloy materials.

General comment: The authors revised their work which now seems to be better organized and logically consistent. However, some further points should be improved:

1) The clarity of the Methods section, where all formulas have been described, should be enhanced, clearly explaining the meaning of the all the used variables.

2) The background of all figures should be white to allow the interested readers to better understand the content of each figure. Please keep attention to the labels to clearly communicate the meaning of the quantities within plots.

3) The references are mainly regional, please extend to other counties .. !

Comments on the Quality of English Language

The quality of English could be further improved, even if, it seems that the current version of the work could be clearly understood.
